# Applicability of a Material Constitutive Model Based on a Transversely Isotropic Behaviour for the Prediction of the Mechanical Performance of Multi Jet Fusion Printed Polyamide 12 Parts

**DOI:** 10.3390/polym16010056

**Published:** 2023-12-23

**Authors:** Sergio Perez-Barcenilla, Xabier Cearsolo, Amaia Aramburu, Ruben Castano-Alvarez, Juan R. Castillo, Jorge Gayoso Lopez

**Affiliations:** 1TECNALIA, Basque Research and Technology Alliance (BRTA), Astondo Bidea, Edificio 700, 48160 Derio, Spain; ruben.castano@tecnalia.com; 2IMH Campus, Azkue Auzoa 1, 20870 Elgoibar, Spain; cearsolo@imh.eus (X.C.); j_r_castillo@imh.eus (J.R.C.); 3TECNALIA, Basque Research and Technology Alliance (BRTA), Mikeletegi Pasealekua 2, 20009 Donostia-San Sebastián, Spain; amaia.aramburu@tecnalia.com (A.A.); jorge.gayoso@tecnalia.com (J.G.L.)

**Keywords:** additive manufacturing, multi jet fusion, polyamide 12, anisotropy, mechanical properties, material constitutive models

## Abstract

Multi Jet Fusion (MJF), an innovative additive manufacturing (AM) technique in the field of Powder Bed Fusion (PBF) developed by Hewlett-Packard (HP) Inc. (Palo Alto, CA, USA), has been designed to produce polymer parts using thermoplastic-based powders, primarily focusing on polyamide 12 (PA12). Employing a layer-by-layer approach, MJF enables the rapid production of intricate components, reportedly up to 10 times faster than other AM processes. While the mechanical properties of MJF-printed PA12 and the impact of build orientation on those properties have already been explored in various studies, less attention has been given to the mechanical performance of MJF-printed PA12 components under complex loads and accurate predictive models. This contribution aims to assess the applicability of a constitutive model based on a transversely isotropic behaviour under linear elastic deformation for predicting the mechanical response of MJF-printed PA12 parts through numerical simulations. Both uniaxial tensile and shear tests were carried out on printed samples to determine the elastic properties of MJF-printed PA12, with additional testing on printed complex handle-shaped parts. Finally, a numerical model was developed to simulate the mechanical tests of the handles. Results from tests on printed samples showed that MJF-printed PA12, to some extent, behaves as a transversely isotropic material. Furthermore, using a constitutive model that assumes a transversely isotropic behaviour under linear elastic deformation for predicting the mechanical response of MJF-printed PA12 parts in numerical simulations could be a reasonable approach, provided that the material stress levels remain within the linear range. However, the particularities of the stress-strain curve of MJF-printed PA12 complicate determining the elasticity-to-plasticity transition point.

## 1. Introduction

ISO/ASTM 52900 [1] provides a definition for additive manufacturing (AM), describing it as the “process of joining materials to make parts from 3D model data, usually layer upon layer, as opposed to subtractive and formative manufacturing methodologies”. AM technologies are categorised in that standard into seven distinct groups, with Powder Bed Fusion (PBF) being one of them. Specifically, PBF involves processes that utilise thermal energy for selectively fusing specific areas within a powder bed. One of the main advantages of PBF techniques is their high resolution. Additionally, the unsolidified powder within the bed serves as a support for the part being printed during the manufacturing process, allowing for the precise fabrication of intricate components, including those with significant overhangs [2].

The two most prominent PBF technologies for manufacturing polymer parts are Selective Laser Sintering (SLS) and Multi Jet Fusion (MJF). The latter was developed and patented by Hewlett-Packard (HP) Inc. While both technologies are capable of delivering reproducible and high-quality results [3], MJF, introduced in 2014 [4], can produce parts of higher density and lower porosity compared to SLS [5]. Moreover, according to HP Inc., MJF can achieve production speeds up to 10 times faster than other additive processes, such as Fused Deposition Modeling (FDM) or SLS [6].

Figure 1 schematically illustrates the various phases of the MJF printing process. In this process, printing begins with the deposition of a thin layer (80 µm) of powder within the build chamber (Figure 1a). Subsequently, energy is applied to this fresh layer to control the temperature of the material immediately before printing agents are added (Figure 1b). Once this preparatory step is completed, a fusing agent (which is black and capable of absorbing infrared energy) is selectively printed at the locations of the particles to be fused together (Figure 1c). In addition, a detailing agent (which is transparent) is selectively applied where the fusing action needs to be either minimised or intensified (Figure 1d). The role of the fusing agent is to facilitate the fusion between the powder particles that it impregnates, while the detailing agent regulates the heat transfer to the powder that has not been impregnated with the fusing agent. This fine-tuning allows for the production of printed parts with a well-defined perimeter and high quality. Finally, the build chamber is exposed to heat using infrared lamps (Figure 1e). This radiation is absorbed by the fusing agent, converting it into thermal energy, thereby intensifying the fusion of the powder particles impregnated with the fusing agent (Figure 1f). This process is repeated layer by layer until the part is completely manufactured.

As the detailing agent limits the heat transfer beyond the boundaries of the component that is being printed, the unfused powder is less degraded and can be reused in subsequent printing processes [7]. Typically, the powder used in MJF printing processes consists of approximately 20% virgin powder and 80% recycled powder.

Polyamide 12 (PA12) is the predominant printing material in MJF processes [7,8,9,10,11,12] because its melting temperature is significantly higher than its crystallisation temperature [13]. In addition to PA12, MJF technology can also utilise other printing materials such as polyamide 11 (PA11), polyamide 12 with glass beads (PA12 GB), and thermoplastic polyamide elastomers (TPA) [14].

Given that MJF is a relatively recent printing technique, there is limited scientific literature available on the subject to date [5,7,15]. Regarding the anisotropy of the mechanical properties of PA12 printed with MJF technology, the first published research is attributed to O’Connor et al. [13]. Results from tests carried out in that work pointed to an isotropic behaviour of the printed material in terms of tensile strength and a notable impact of the build orientation on the flexural properties of the MJF-printed PA12. The authors concluded that the mechanical performance of MJF-printed PA12 was comparable to that of other production processes based on PBF technology.

That study was followed by another work conducted by O’Connor and Dowling [16], in which the mechanical performance of two MJF-printed materials, PA12 and PA12 GB, were compared. Regarding the impact of the printing orientation on the mechanical properties of the printed materials, tensile test results pointed to an isotropic behaviour for both PA12 and PA12 GB in terms of strength and stiffness. However, flexural tests did reveal a degree of anisotropy in the mechanical properties of both printed materials. The authors attributed the anisotropy observed in the flexural behaviour of the printed materials to the influence of the build orientation on the material porosity, which affected flexural mechanical properties more than tensile mechanical properties.

Morales-Planas et al. [17] also studied the mechanical behaviour of MJF-printed PA12. According to the authors, the results obtained from tensile tests on printed samples yielded no clear correlation between build orientation and maximum tensile stress.

In contrast, Palma et al. [5] did observe a certain influence of the printing orientation on the mechanical properties of MJF-printed PA12. Tensile test results confirmed lower fracture strain and slightly higher fracture stress within the vertically oriented samples compared to the horizontally oriented ones. According to the authors, those results might be attributed to the layer interfaces and their orientation with respect to the direction of the applied load during the tensile tests and to the fact that the interfaces were notably stronger, though also more brittle.

Riedelbauch et al. [15] assessed the impact of the ageing of PA12 powders and the mixing ratio (virgin powder/aged powder) on the mechanical properties of MJF-printed material. The authors observed that the vertically oriented printed samples showed better tensile performance when compared to the horizontally oriented ones and attributed that behaviour to the additional layer weight with increasing Z-height added by the fusing agent, leading to printed parts of higher density and lower porosity along the Z-axis.

Galati et al. [9] performed tensile tests on specimens taken from an industrial component and on standardised samples, which were fabricated with the same orientation in the build chamber as the extracted specimens. Both the industrial component and the samples were manufactured using the same MJF printer and material from the same batch. Although similar tensile strength results for the same build orientation were noted for the specimens from the printed industrial component and for the printed samples, the printed samples displayed significantly higher deformation at break. According to the authors, that difference can be attributed to the increased likelihood of defects being induced in real parts with complex geometries, which can have a detrimental effect on the deformation at break results. Consequently, the authors concluded that standardised tensile samples can reliably estimate the strength of real end-products printed using MJF technology, but, on the other hand, they may overestimate the deformation at break of those real end-products.

Sillani et al. [18] conducted a comparison of the tensile mechanical properties of both SLS and MJF-printed samples utilising the same printing material, PA12. In the case of MJF technology, the vertically oriented samples yielded higher strength and stiffness. According to the authors, that behaviour can be partially explained by higher densification at the interlayer boundary due to the action of the fusing agent, which allows the creation of stronger interlayer bonding for smaller cross-sections.

Mehdipour et al. [19] also conducted a comparative analysis of the tensile mechanical properties of PA12 samples produced using both SLS and MJF printing processes. Consistent with earlier findings, in the case of MJF technology, the vertically oriented samples (referred to as “upright” in that work) exhibited the most favourable performance. Furthermore, in addition to the anisotropy of the mechanical properties of MJF-printed PA12, the authors noted that the test speed also influenced the mechanical performance of the printed material.

A comparison of the mechanical properties between SLS and MJF-printed PA12 parts was also explored by Cai et al. [8]. Their study revealed a degree of anisotropy in the mechanical properties of the MJF-printed material, with vertically oriented samples displaying the best tensile performance. The authors concluded that MJF-printed parts exhibited strong bonding between layers, primarily attributed to the presence of carbon black in the fusing agent, which enhanced interlayer integrity. Moreover, the addition of the fusing agent increased the density and reduced the porosity of the printed part along the vertical direction, as Riedelbauch et al. [15] also noted.

Another study comparing SLS and MJF printing technologies was carried out by Rosso et al. [20]. In that work, the authors did not specifically analyse the influence of the printing orientation on the mechanical properties of the printed material. However, as a valuable contribution, the authors compiled results from the literature on the tensile properties of MJF-printed PA12 parts and compared them with their own findings.

The works analysed so far addressed the anisotropy of MJF-printed PA12 parts. In those studies, the build orientation was defined in terms of the orientation of the principal axis of the samples relative to the printer reference system. However, Calignano et al. [7] adopted a more comprehensive approach by investigating the impact of build orientation (determined not only by the orientation of the principal axis of the samples relative to the printer reference system but also by the rotation of the sample around that axis) on the tensile mechanical properties of MJF-printed samples using PA12 as the printing material. Results demonstrated that both parameters (orientation and rotation) influenced the tensile properties of MJF-printed PA12. Once again, the vertically oriented samples were, in general, observed to have the most favourable tensile properties.

Chen et al. [21] focused their efforts on the development of a finite-strain viscoelastic-viscoplastic constitutive model to represent the behaviour of MJF-printed PA12. The results from the tensile tests carried out in the frame of that work indicated that the mechanical properties of the printed material were almost independent of the building angle within the same building plane. However, different tensile properties along the Z-axis compared to the X and the Y-axis were observed in the printed material. Those results were attributed to the layered structure of the printed material resulting from the MJF printing process.

In a subsequent study, Chen et al. [22] assessed the impact of the build orientation on the tensile properties, tension–tension low cycle fatigue behaviour, and failure mechanism of MJF-printed PA12 parts. Tensile test results showed similar strengths for both vertical and horizontal orientations. However, a slightly higher Young’s modulus and a significantly lower elongation at break were noted in the vertically oriented samples. The authors attributed the increased stiffness of the vertically oriented samples to the carbon black-enhanced sintering interfaces. Furthermore, the reduced elongation at break in the vertically oriented samples was attributed to the brittle features of the sintering interfaces and to the higher presence of void defects perpendicular to the loading direction.

Osswald et al. [23] established a criterion for predicting the failure of MJF-printed PA12 components subjected to complex stress states. As an initial hypothesis, the authors assumed that the mechanical properties of components produced using printing technologies such as SLS and MJF, due to their layered structure and layer generation processes, corresponded to those of transversely isotropic material. To calibrate that failure criterion, the authors conducted a series of tests on non-standard cylindrical specimens. The test results revealed significant tensile and compressive strength disparities. Additionally, the vertically oriented samples showed higher tensile and compressive strengths compared to the horizontally oriented ones.

Abdallah et al. [24] conducted a study to investigate the influence of printing orientation and strain rate on the tensile properties of MJF-printed PA12. Test results indicated that, at higher strain rates, the specimens printed at the 25° orientation were stiffer than those printed at the 0° orientation. Furthermore, regardless of the strain rate, the elongation at break of the specimens printed at the 25° orientation compared to the specimens printed at the 0° orientation was lower.

Finally, Koh et al. [12] assessed the impact of ageing on the tensile mechanical properties of MJF-printed PA12 parts. Test results after one day of storage indicated that the X-axis had the highest values for both tensile strength and elongation at break, while the tensile modulus values were similar across the three different build orientations that were analysed.

In addition to build orientation, the location of the printed parts within the build chamber is another factor that can influence the mechanical properties of MJF-printed PA12, as reported by Chen et al. [25]. The authors observed that specimens located in the central areas of the build chamber exhibited less elongation at break compared to those positioned at the perimeter, suggesting that differences in the position of the printed samples within the build chamber could explain the variability in the mechanical properties of MJF-printed PA12 reported in the literature.

Table 1 presents a comprehensive overview of all the above-mentioned works. For each of them, pertinent details are shown, encompassing the specific tests conducted, the printer model and the print mode employed for sample fabrication, and the build orientations considered during the sample printing process.

Considering the conclusions drawn from the analysis of all those works, the following key points can be highlighted with regard to MJF-printed PA12:In all the studies where the printing mode for manufacturing the samples/parts was specified, the Balanced Print Mode was consistently employed. This printing mode offers an optimal balance between printing speed and the final properties (mechanical properties, dimensional accuracy, appearance, surface quality, and more) of the printed parts [13,15,16,21].To evaluate the tensile properties of MJF-printed PA12, two different standards are applicable: ISO 527, as referenced in [7,9,13,15,18,19,20], and ASTM D638, as referenced in [5,8,12,16,17,21,24]. It should be noted that the tensile properties of MJF-printed PA12 listed in the catalogues published by the developer of MJF technology (HP Inc.) [14] were determined in reference to ASTM D638. Similarly, the procedures specified in both ISO 178 (see [8,13]) and ASTM D790 (see [16]) are suitable for assessing the flexural properties of MJF-printed PA12.The layered structure derived from the MJF printing technology and the method used to generate those layers suggest a mechanical behaviour of the MJF-printed PA12 that is analogous to that of a transversely isotropic material. Transversely isotropic materials exhibit an isotropic behaviour within the plane of isotropy, with mechanical properties that differ from those in a direction perpendicular to that plane. In general, higher values for strength and stiffness were observed in the growing direction compared to the printing plane, along with lower values for elongation at break [5,7,8,13,15,18,19,22,23]. This observation is consistent with the data provided by HP Inc. [14] regarding the mechanical properties of MJF-printed PA12.Despite the consistent use of the same printer model (HP Jet Fusion 4200) in most of the studies that were analysed [7,8,9,13,15,16,17,18,19,20,23] for the manufacture of the specimens, significant dispersion exists in the reported results for the mechanical properties of MJF-printed PA12.Overall, the mechanical properties of MJF-printed PA12 reported in the various studies that were analysed tend, to varying degrees, to have lower values than those published by HP Inc. [14].

The aim of this study is to assess the applicability of a material constitutive model based on a transversely isotropic behaviour under linear elastic deformation for predicting, by means of numerical calculations, the mechanical response of MJF-printed PA12 parts. First, assuming that the MJF-printed PA12 behaves as a transversely isotropic material, its elastic parameters (elastic moduli, shear moduli, and Poisson’s ratios) will be determined through both uniaxial tensile and shear tests on samples printed along four different build orientations (YX, XY, ZY, and ZX). Both the YX and the XY orientations, which lie on the printing plane, are theoretically equivalent. Moreover, both the ZY and the ZX orientations, which are perpendicular to the printing plane, can also be considered theoretically equivalent. The comparison of the test results from samples printed in theoretically equivalent orientations provided conclusions on the transversely isotropic behaviour of MJF-printed PA12. Finally, three handles with different build orientations were manufactured using MJF printing technology and PA12 as the printing material for mechanical testing. A numerical model, including a material constitutive model based on a transversely isotropic behaviour under linear elastic deformation, was also developed to simulate the mechanical tests conducted on the printed handles. Results from both the mechanical tests and the numerical model were then compared in order to assess the applicability of the selected material constitutive model to predict the mechanical response of MJF-printed PA12 parts.

## 2. Materials and Methods

### 2.1. Constitutive Model for Transversely Isotropic Materials

If numerical simulations are used to predict the structural response of components, then a constitutive model that can accurately represent the behaviour of the component material is essential. Considering the particularities of the MJF printing technology, a transversely isotropic behaviour was assumed in this work for MJF-printed PA12. Orthotropic materials are characterised by having three mutually perpendicular planes of symmetry. Transversely isotropic materials are a specific subset of orthotropic materials in which one of those three planes of symmetry is also isotropic.

Equation (1), derived from the generalised Hooke’s Law, characterises under elastic and linear conditions where the material experiences small deformation and assumes equivalent behaviours under both tension and compression, the strain-stress relationship of transversely isotropic materials whose plane of isotropy is parallel to the XY plane. In the case of MJF-printed PA12, the plane of isotropy coincides with the printing plane, whereas the growing direction of the part being printed aligns with the Z-axis.
(1)$$\left(\begin{array}{c} \varepsilon_{x}\\ \varepsilon_{y}\\ \varepsilon_{z}\\ \gamma_{yz}\\ \gamma_{xz}\\ \gamma_{xy} \end{array}\right)=\left(\begin{array}{cccccc} \begin{array}{c} 1/E_{1}\\ -\nu_{1}/E_{1}\\ -\nu_{2}/E_{2}\\ 0\\ 0\\ 0 \end{array} & \begin{array}{c} -\nu_{1}/E_{1}\\ 1/E_{1}\\ -\nu_{2}/E_{2}\\ 0\\ 0\\ 0 \end{array} & \begin{array}{c} -\nu_{2}/E_{2}\\ -\nu_{2}/E_{2}\\ 1/E_{2}\\ 0\\ 0\\ 0 \end{array} & \begin{array}{c} 0\\ 0\\ 0\\ 1/G_{2}\\ 0\\ 0 \end{array} & \begin{array}{c} 0\\ 0\\ 0\\ 0\\ 1/G_{2}\\ 0 \end{array} & \begin{array}{c} 0\\ 0\\ 0\\ 0\\ 0\\ 1/G_{1} \end{array} \end{array}\right)\left(\begin{array}{c} \sigma_{x}\\ \sigma_{y}\\ \sigma_{z}\\ \tau_{yz}\\ \tau_{xz}\\ \tau_{xy} \end{array}\right)$$

In Equation (1), $\varepsilon_{i}$ is the longitudinal strain in the i direction; $\gamma_{ij}$ is the shear strain on the ij plane; $\sigma_{i}$ is the longitudinal stress in the i direction; and $\tau_{ij}$ is the shear stress on the ij plane. It is fulfilled that:(2)$$E_{1}=E_{x}=E_{y}\;E_{2}=E_{z}\;G_{1}=G_{xy}=E_{1}/\left(2\cdot \left(1+\nu_{1}\right)\right)\;G_{2}=G_{yz}=G_{xz}\;\nu_{1}=\nu_{xy}=\nu_{yx}\;\nu_{2}=\nu_{xz}=\nu_{yz}$$
where $E_{i}$ is the elastic (Young’s) modulus in the i direction; $\nu_{ij}$ is the Poisson’s ratio, representing the contraction along the i axis when an elongation along the j axis occurs; and $G_{ij}$ is the shear modulus on the ij plane.

Thus, five parameters must be determined in order to fully define the flexibility matrix that characterises the linear elastic behaviour of a transversely isotropic material: two elastic moduli ($E_{1}$ and $E_{2}$), two Poisson’s ratios ($\nu_{1}$ and $\nu_{2}$), and one shear modulus ($G_{2}$). Simplifying the problem of determining those parameters, the overall three-dimensional behaviour can be decomposed into two simpler plane stress cases: one plane stress case on the XY plane, and the other on either the XZ or the YZ plane.

Assuming a plane stress state on the XY plane:(3)$$\sigma_{z}=\tau_{yz}=\tau_{xz}=0$$

Then, the strain-stress relationship for a transversely isotropic material under a plane stress state on the XY plane can be represented as follows:(4)$$\left(\begin{array}{c} \varepsilon_{x}\\ \varepsilon_{y}\\ \gamma_{xy} \end{array}\right)=\left(\begin{array}{c} 1/E_{1}\\ -\nu_{1}/E_{1}\\ 0 \end{array}\;\begin{array}{c} -\nu_{1}/E_{1}\\ 1/E_{1}\\ 0 \end{array}\;\begin{array}{c} 0\\ 0\\ 1/G_{1} \end{array}\right)\left(\begin{array}{c} \sigma_{x}\\ \sigma_{y}\\ \tau_{xy} \end{array}\right)$$
where:(5)$$G_{1}=E_{1}/\left(2\cdot \left(1+\nu_{1}\right)\right)$$

According to Equations (4) and (5), only two elastic parameters ($E_{1}$ and $\nu_{1}$) are required to define the linear elastic behaviour of a transversely isotropic material under a plane stress state on its plane of isotropy (XY plane). In the initial phase of a uniaxial tensile test, when the material remains within its linear elastic range, it can be assumed that the sample will behave as if it were in a plane stress state. In this scenario, those parameters ($E_{1}$ and $\nu_{1}$) can be determined through a tensile test performed on a single sample oriented in any direction on the XY plane.

It is important to note that a plane stress state on the XY plane will not imply zero longitudinal strain along the Z-axis. In that scenario, while $\gamma_{yz}=\gamma_{xz}=0$, the longitudinal strain along the Z-axis can be determined using Equation (6):(6)$$\varepsilon_{z}=-\nu_{2}\cdot \left(\sigma_{x}+\sigma_{y}\right)/E_{2}$$

Hence, Equation (6) can be used to calculate the out-of-plane strain induced by in-plane loads.

Similarly, assuming a plane stress state on the XZ plane:(7)$$\sigma_{y}=\tau_{yz}=\tau_{xy}=0$$
then:(8)$$\left(\begin{array}{c} \varepsilon_{x}\\ \varepsilon_{z}\\ \gamma_{xz} \end{array}\right)=\left(\begin{array}{c} 1/E_{1}\\ -\nu_{2}/E_{2}\\ 0 \end{array}\;\begin{array}{c} -\nu_{2}/E_{2}\\ 1/E_{2}\\ 0 \end{array}\;\begin{array}{c} 0\\ 0\\ 1/G_{2} \end{array}\right)\left(\begin{array}{c} \sigma_{x}\\ \sigma_{z}\\ \tau_{xz} \end{array}\right)$$

In the above scenario, in addition to determine $E_{1}$, three additional elastic parameters ($E_{2}$, $\nu_{2}$, and $G_{2}$) must be determined, in order to characterize the linear elastic behaviour of a transversely isotropic material under plane stress conditions on the XZ plane. $E_{2}$ and $\nu_{2}$ can be derived through a tensile test performed on a single sample oriented along the Z-axis. Meanwhile, $G_{2}$ can be calculated through a shear test on a sample oriented along the Z-axis.

An alternative approach to determine $G_{2}$ is to combine the results from the uniaxial tensile tests on the samples oriented along both the X-axis (or any other direction on the XY plane) and the Z-axis with the results of a third uniaxial tensile test on a sample at an angle of 45° in relation to both the X and the Z-axis, using Equation (9).
(9)$$G_{2}=\frac{1}{\frac{4}{E_{45{}^{\circ}}}-\frac{1}{E_{1}}-\frac{1}{E_{2}}+\frac{2\cdot \nu_{2}}{E_{2}}}\;$$

If a plane stress state on the YZ plane (rather than the XZ plane) is assumed, then the same procedure will yield equivalent results.

This approach provides a comprehensive characterisation of the behaviour of the MJF-printed PA12 within its linear elastic range by decomposing its overall three-dimensional behaviour into two-plane stress states, which are simpler scenarios.

### 2.2. Samples/Parts Fabrication

In this study, an HP Jet Fusion 4200 3D printer equipped with MJF technology was used to produce the samples and parts for testing, using PA12 powder as the printing material. Table 2 provides a summary of the key printing process parameters and the primary properties of the PA12 powder.

In all cases, the Balanced Print Mode was used for printing. Following the recommendations of the printer manufacturer (HP Inc.), the samples/parts were cooled off naturally within the build chamber after the printing process. After removal from the build chamber, air blasting eliminated any residual surface powder.

#### 2.2.1. Manufacture of Tensile and Shear Samples

The samples for uniaxial tensile and shear tests were produced in a single printing operation. Four different build orientations were used: YX, XY, ZY, and ZX. The first letter denotes the orientation of the primary dimension (length) of the sample, while the second indicates the orientation of the secondary dimension (width) relative to the printer reference system. In that context, X represents the direction of the printhead’s movement, Y is the recoating direction, and Z is the vertical direction (the growing direction of the part being printed). Figure 2 illustrates the layout of the specimens for uniaxial tensile and shear tests within the build chamber of the printer. The location of the samples within the build chamber was randomly determined.

A total of 40 samples were printed, comprising 5 tensile and 5 shear samples for each of the 4 build orientations that were analysed. This breakdown consisted of 20 samples designated for uniaxial tensile testing and another 20 samples intended for shear testing.

For uniaxial tensile testing, two different types of samples, type 1A (injection moulded) and type 1B (machined), are specified in ISO 527-2 [26]. In this work, as the samples were printed to their final dimensions, type 1A samples were chosen; the geometry and dimensions are illustrated in Figure 3. Figure 4 shows the printed samples prior to submitting them to uniaxial tensile testing.

On the other hand, the geometry of the shear samples was chosen according to the recommendations outlined in ASTM D5379/5379M [31]. The final dimensions of those samples (see Figure 5) were as follows: $d_{1}=19.0\;mm$, $d_{2}=3.8\;mm$, L=76.0 mm, r=1.3 mm, and w=11.4 mm. Regarding the thickness of the samples, h, ASTM D5379/5379M affords some flexibility when selecting the most suitable value. In this work, a thickness value of 10 mm was chosen to minimise the risk of sample buckling during shear testing.

Shear testing requires precise parallel alignment during the test between the long faces of the sample that are in contact with the testing device. In view of that requirement, oversized prismatic specimens measuring 23 × 76 × 10 mm were initially printed (see Figure 6). Those specimens were then machined to their final dimensions, and notches were subsequently added after the printing process.

#### 2.2.2. Manufacture of Handles for Mechanical Testing

The geometry and dimensions of the handles analysed in this study were identical to those used by Domingo-Espín et al. [32] in their work (see Figure 7). The handles were intentionally designed to induce a complex stress state during testing [32]. They consisted of two perpendicular straight arms, each with a rectangular cross-section measuring 10 × 15 mm. Additionally, the handles featured a square base measuring 60 × 60 × 5 mm, which served to secure them to the testing device.

As the MJF-printed PA12 was supposed to exhibit a transversely isotropic behaviour, the printing orientations for the handles, determined by the face of the handle in contact with the printing bed, were narrowed down to just three options. In this case, faces 1, 2, and 5 (see Figure 7) were selected. For each of the three selected printing orientations, 1 handle was printed, resulting in a total of 3 printed handles, namely 1-H, 2-H, and 5-H. Figure 8 illustrates the layout of the handles for mechanical testing within the build chamber of the printer. The location of the handles within the build chamber was randomly determined.

### 2.3. Mechanical Testing

#### 2.3.1. Uniaxial Tensile Tests

Uniaxial tensile tests following ISO 527-1 [33] were performed to determine the tensile elastic parameters that will characterise the behaviour of the MJF-printed PA12 within the linear elastic range, assuming a transversely isotropic behaviour. Those parameters included the elastic modulus on the plane of isotropy ($E_{1}=E_{x}=E_{y}$), the elastic modulus perpendicular to the plane of isotropy ($E_{2}=E_{z}$), the Poisson’s ratio on the plane of isotropy ($\nu_{1}=\nu_{xy}$), and the Poisson’s ratio on a plane perpendicular to the plane of isotropy ($\nu_{2}=\nu_{xz}=\nu_{yz}$). Moreover, the tensile strengths of the MJF-printed PA12, both on and perpendicular to the plane of isotropy, were also determined by means of these tests.

An INSTRON 5500R universal testing machine equipped with a 100 kN load cell was used for the uniaxial tensile tests. Although 5 samples were manufactured for each of the four build orientations that were analysed, only three of them were subjected to testing, with the remaining two held in reserve. Consequently, a total of 12 samples underwent uniaxial tensile testing. One of the three tested samples was instrumented to measure the elastic modulus, while the other two were instrumented to measure both the elastic modulus and the Poisson’s ratio for each of the build orientations. Data acquisition was consistently performed using a StrainSmart 8000 system, with a data acquisition frequency set at 10 Hz.

Each of the tested samples underwent two tests. Initially, to determine the elastic modulus (and, when applicable, the Poisson’s ratio), the samples were tested at a strain rate of 1 mm/min until a strain between 0.3% and 0.5% was achieved, after which the samples were unloaded. The elastic modulus, as defined in ISO 527-1, was calculated as the slope of the stress-strain curve within the range of 0.05% to 0.25% strain. On the one hand, strain levels were recorded during the test using a contact extensometer with a reference length, $L_{o}$, set at 50 mm (see Figure 3) to measure the elastic modulus. On the other hand, the Poisson’s ratio was calculated as the negative ratio between the change in deformation in the direction corresponding to the width of the sample (transverse strain) and the change in deformation in the lengthwise direction (longitudinal strain). Similar to the elastic modulus, the Poisson’s ratio was determined within the strain range of 0.05% and 0.25%. The samples were equipped with biaxial strain gauges for the determination of the Poisson’s ratio.

After completing the initial test, the samples underwent a second test up until failure, at a strain rate of 5 mm/min, yielding stress-strain curves. Stress values were determined using the area of the initial cross-section of the sample, while strain was calculated based on the displacement of the grips of the testing machine, using an initial grip distance value, L, of 115 mm, in accordance with ISO 527-2. Finally, tensile strength was computed as the maximum test load divided by the area of the initial cross-section of the sample.

#### 2.3.2. Shear Tests

Shear tests, as specified in ASTM D5379/D5379M, were performed to determine the shear elastic parameters that will characterise the behaviour of the MJF-printed PA12 within the linear elastic range, assuming a transversely isotropic behaviour. Those parameters included the shear modulus on the plane of isotropy ($G_{1}=G_{xy}$) and the shear modulus on a plane perpendicular to the plane of isotropy ($G_{2}=G_{yz}=G_{xz}$). In addition, the same tests were used to determine the shear strengths of the MJF-printed PA12 both on and perpendicular to the plane of isotropy.

Similar to the uniaxial tensile tests, shear tests were conducted using an INSTRON 5500R universal testing machine equipped with a 100 kN load cell. Although 5 samples were manufactured for each of the four build orientations that were analysed, only three of them were tested, with the remaining two held in reserve. Consequently, a total of 12 samples were subjected to shear testing. All samples were tested at a strain rate of 2 mm/min in accordance with ASTM D5379/D5379M recommendations, and they were all equipped with biaxial strain gauges. Data acquisition was consistently performed using a StrainSmart 8000 system, with a data acquisition frequency set at 10 Hz. Figure 9 shows one of the shear samples mounted on the testing device.

For each of the tested samples, shear stress was determined using the initial cross-sectional area of the sample at the notch location. Additionally, the shear strain was determined based on the strain gauge measurements. The shear modulus was calculated as the chord modulus within the strain range of 1500 to 5500 µε, following the recommendations specified in ASTM D5379/D5379M. Finally, shear strength was calculated by taking the lower value between the maximum test load and the load corresponding to a 5% strain, as specified in ASTM D5379/D5379M.

### 2.4. Mechanical Tests on Printed Handles

The mechanical tests on the printed handles were conducted using an INSTRON 5500R universal testing machine equipped with a 10 kN load cell. The experimental test setup is illustrated in Figure 10. As shown in that figure, the 60 × 60 × 5 mm square base of the printed handles was secured to the testing machine fixture, forming a cantilever. Using a cylindrical-shaped component, a controlled vertical displacement (50 mm, at a strain rate of 5 mm/min) was applied to the upper surface of the handle, specifically along the line corresponding to the cross-section initially located at 15 mm from its edge, with the testing device providing the necessary load to induce the displacement.

## 3. Results and Discussion

### 3.1. Uniaxial Tensile Tests

The results of the uniaxial tensile tests are presented in Figure 11 and Table 3. Since it was assumed in this study a mechanical behaviour consistent with that of a transversely isotropic material for the MJF-printed PA12, theoretically equivalent build orientations are grouped together in Table 3.

On the basis of the results presented in Table 3, it can be concluded that the parts manufactured through MJF printing technology using PA12 as the printing material had superior tensile properties along the vertical direction (ZY and ZX orientations) when compared to the horizontal printing plane (YX and XY orientations). In particular, the printed material exhibited tensile strength values and elastic moduli along the vertical direction that were approximately 16% greater than those observed on the printing plane. The results also pointed to relatively consistent Poisson’s ratio values on both the plane of isotropy (YX and XY orientations) and the planes that were perpendicular to the plane of isotropy (ZY and ZX orientations). Finally, the significant variability of the results yielded no definitive conclusions with which to determine the impact of the build orientation on elongation at break.

As previously stated, the anisotropic nature of the tensile properties of the MJF-printed PA12, superior along the vertical direction in comparison to the horizontal plane, was noted by several authors [5,7,8,13,15,18,19,22,23]. They primarily attributed this phenomenon to the presence of the fusing agent, which enhances adhesion, resulting in stiffer and stronger bonds between layers of fused material [7,8,15,18,22]. However, the addition of the fusing agent also leads to a more brittle behaviour of the printed material, characterised by a reduced deformation capability [8].

The results of the tensile tests revealed a substantial dispersion in terms of tensile strength and, especially, elongation at break, much higher than that observed in other works on the mechanical characterisation of the MJF-printed PA12 [7,8,13,18,19,20,22,23]. This scatter of the results significantly lessened in the case of the elastic modulus, and notably so in the case of the Poisson’s ratio.

The results of this study with regard to the tensile mechanical properties of MJF-printed PA12 revealed lower values for both the tensile strength and the elastic modulus, as well as elongation at break when compared to the data supplied by HP Inc. [14].

Table 4 summarises published data regarding the tensile properties of MJF-printed PA12.

### 3.2. Shear Tests

The shear test results are presented in Figure 12 and Table 5. Since it was assumed in this study a mechanical behaviour consistent with that of a transversely isotropic material for the MJF-printed PA12, theoretically equivalent build orientations are grouped together in Table 5.

As observed in Figure 12, the ZX-3-S sample exhibited unexpected behaviour during the shear test. Consequently, data from that sample were excluded from the calculations in Table 5 for determining the average shear properties of the MJF-printed PA12 on the planes perpendicular to the plane of isotropy.

The shear properties of the MJF-printed PA12 showed a noticeable reduction in anisotropy when compared to the tensile properties. Almost identical shear modulus values were observed on both the plane of isotropy (YX and XY orientations) and the vertical planes (ZY and ZX orientations). Slightly higher shear strength values (6.7% increase) were also recorded on the vertical planes.

### 3.3. Transversely Isotropic Behaviour of the MJF-Printed PA12

The hypothesis considered in this work based on the transversely isotropic behaviour of the MJF-printed PA12 allowed a combination of the results from tests on the YX and the XY-oriented printed samples for determining the mechanical properties of the printed material on the plane of isotropy and the results of the tests on the ZY and the ZX oriented printed samples, for estimating the mechanical properties of the printed material along the growing direction (see Table 3 and Table 5).

Thus, the comparison of the test results on samples printed along theoretically equivalent orientations (YX orientation vs. XY orientation and ZY orientation vs. ZX orientation) led to conclusions regarding the transversely isotropic behaviour of the MJF-printed PA12. Figure 13, Figure 14, Figure 15, Figure 16 and Figure 17, respectively, show the results related to tensile strength, elastic modulus, Poisson’s ratio, shear strength, and shear modulus of the tested samples, grouped by the different build orientations (YX, XY, ZY and ZX) that were studied.

Considering the data depicted in Figure 13, Figure 14, Figure 15, Figure 16 and Figure 17, all the parameters under analysis showed very similar average values along both the ZY and the ZX orientations, with the maximum difference between orientations being approximately 2%. Regarding the printing plane, the discrepancy in average values between theoretically equivalent build orientations (YX and XY orientations) was more pronounced compared to the difference between the ZY and the ZX orientations. Specifically, disparities of 17.42%, 15.05%, 7.89%, and 7.91% were observed for tensile strength, elastic modulus, shear strength, and shear modulus, respectively. In all cases, the mechanical properties related to the YX orientation were superior to those related to the XY orientation. Finally, concerning Poisson’s ratio, the average values obtained from tests were quite similar in both the YX and the XY orientations, with only a 1.90% difference. Taking all the above into account, the assumption of transversely isotropic material behaviour for the MJF-printed PA12 can be considered to be a reasonably valid approach.

### 3.4. Structural Behaviour of Handles

With regard to the mechanical tests on the printed handles, the curves illustrating the time-dependent evolution of the load to be applied by the testing device to generate the defined displacement profile (maximum vertical displacement of 50 mm, at a strain rate of 5 mm/min) of each test specimen (1-H, 2-H and 5-H), are shown in Figure 18. A significant similarity among all three handles can be observed in this figure, with handle 5-H displaying a slightly greater stiffness (+6% approximately) compared to handles 1-H and 2-H.

A numerical model was also developed in this study to simulate the mechanical tests conducted on the printed handles. The finite element analysis software ANSYS 2023R1 was used to create the numerical model, whose geometry is depicted in Figure 19. For the sake of simplicity, the 60 × 60 × 5 mm square base of the handle was not included in the numerical model. Instead, the displacement of the nodes on the face of the handle that was connected to the square base was constrained.

As shown in Figure 19, the actual conditions of the mechanical tests were reproduced by modelling a cylinder with its longitudinal axis located 15 mm from the edge of the handle. A vertical displacement (50 mm along the -Y-axis) was imposed on that cylinder in 25 substeps, and a frictionless contact was defined between the handle and the cylinder. The presence of that contact required a nonlinear analysis, which was solved using an implicit solver.

Among the material constitutive models available in ANSYS 2023R1, there is no constitutive model based on a transversely isotropic behaviour, such as the one defined in Equation (1). Under these circumstances, a more general material constitutive model, based on an orthotropic behaviour under linear elastic deformation, was employed to represent the behaviour of the MJF-printed PA12. The dialogue box where the properties of this material model must be defined is shown in Figure 20.

In Figure 20, EX, EY, and EZ are the elastic moduli along the X,  Y, and Z axes, GXY, GYZ, and GXZ are the shear moduli on the XY, YZ, and XZ planes, PRXY, PRYZ, and PRXZ are the major Poisson’s ratios on the XY, YZ, and XZ planes, and NUXY, NUYZ, and NUXZ are the minor Poisson’s ratios on the XY, YZ, and XZ planes. ANSYS 2023R1 assumes that the X, Y, and Z axes are perpendicular to the three planes of symmetry of the material. Moreover, the major (PR) and minor (NU) Poisson’s ratios for each of those three planes of symmetry are related through the elastic moduli.
(10)PRXY/EX=NUXY/EYPRXZ/EX=NUXZ/EZPRYZ/EY=NUYZ/EZ

Thus, for each of the planes (XY, YZ, and XZ), it is necessary to define only one of the two Poisson’s ratios. It should be noted that if EX=EY, GYZ=GXZ, PRYZ=PRXZ, and NUYZ=NUXZ; then the orthotropic constitutive model is transformed into a transversely isotropic constitutive model in which the plane of isotropy of the material is parallel to the XY plane.

Considering the results from both uniaxial tensile and shear tests on the MJF-printed PA12 samples (see Table 3 and Table 5) and the build orientation used to print each handle, the material properties employed in the constitutive model available in ANSYS 2023R1 are shown in Table 6 for each of the three printed handles analysed with the numerical model.

Figure 21 displays the outcomes derived from both the numerical model and the mechanical tests concerning the stiffness of the handles. In that figure, the load required to achieve the specified displacement profile is illustrated for both scenarios. Due to convergence issues, the results from the numerical model are depicted in Figure 21 exclusively for displacements of up to 30 mm.

In relation to the results of the numerical model, Figure 21 illustrates the very similar behaviours of all three handles, with handles 1-H and 5-H showing slightly higher stiffness (approximately +3%) compared to handle 2-H. From a qualitative point of view, the results of the numerical model and of the mechanical tests on the printed handles closely resembled each other. However, in quantitative terms, to achieve the same displacement, the load required in the numerical model was between 26% and 32% higher compared to the mechanical tests, although that figure varied between handles.

After testing the printed handles, samples were extracted and subsequently subjected to uniaxial tensile tests, following the procedure described in Section 2.3.1: “Uniaxial tensile tests”, in order to investigate the underlying causes of those differences between results from the numerical model and mechanical tests. Two tensile samples were taken from each of the three tested handles, one from each arm, as illustrated in Figure 22. Given the size of the printed handles, the geometry and dimensions of the extracted samples matched those of sample type 1BA, in accordance with ISO 527-2 (sample type 1B at a scale of 1:2).

The results of the uniaxial tensile tests on the samples extracted from the printed handles are presented in Figure 23 and Table 7. Taking into account the layout of the handles within the build volume of the printer during the manufacturing process (see Figure 8) and assuming that the MJF-printed PA12 behaves as a transversely isotropic material, the results from samples 1-H-A, 2-H-A, 2-H-B, and 5-H-B are grouped in Table 7, as their build orientations are equivalent to the XY and YX orientations. Similarly, the results from samples 1-H-B and 5-H-A are also grouped since their build orientations are equivalent to the ZX and ZY orientations. For the sake of comparison, the results corresponding to the YX + XY and ZY + ZX orientations, derived from the uniaxial tensile tests conducted on MJF-printed PA12 samples (see Table 3), are also included in Table 7.

As shown in Table 7, the values of the elastic moduli employed in the numerical model, derived from uniaxial tensile tests conducted on printed samples, were significantly higher than the uniaxial tensile test results for the samples from the printed handles. In the specific case of the YX + XY orientations, the difference was 18%, whereas for the ZY + ZX orientations, the difference increased to 35%. ISO 527-2 states that the results of the uniaxial tensile testing of the type 1BA samples are not quantitatively comparable to the results of the uniaxial tensile testing of type 1A samples. Nevertheless, the substantial disparity observed in the results of the uniaxial tensile tests performed on both printed samples and samples extracted from the printed handles might, to some extent, help explain the differences between the results of the mechanical test on the printed handles and the results of the numerical model. Differences between test results of both printed samples and samples obtained from a more complex printed part were already detected by Galati et al. [9] and attributed to the increased likelihood of defects being induced in real parts with complex geometries. In this case, as no additional works were carried out to determine the density and distribution of printing defects in both the printed samples and the samples extracted from the printed handles, the findings observed by those authors could not be verified.

Another potential reason for the disparity observed between the results of the mechanical tests on the printed handles and the theoretical predictions of the numerical model when simulating the mechanical response of those handles during the tests may stem from the fact that, as previously mentioned, the constitutive model used in the numerical simulations to characterise the mechanical behaviour of the MJF-printed PA12 was based on a transversely isotropic behaviour under linear elastic deformation. With this model, the values of the elastic moduli remained constant throughout the entire simulation, regardless of the stress/strain level attained by the material. Consequently, as illustrated in Figure 21, the load-displacement curves for the handles obtained from the numerical model can be more or less accurately represented by a straight line. In contrast, in the context of the mechanical tests conducted on the printed handles, taking into account the evolution of the stress-strain relationship for the MJF-printed PA12 shown in Figure 11 and Figure 23, as those tests progress and the stress on the printed material increases, the stiffness of the handles should gradually diminish. That behaviour can be observed in the evolution of the load-displacement curves based on the results of the mechanical tests of the printed handles, shown in Figure 21, tracing a decreasing slope as the displacement increased. Figure 24 illustrates that, during the numerical simulations of the mechanical tests on the printed handles, very high Von Mises equivalent stress values were recorded for the printed material at localised regions (for example, in the case of handle 5-H, a maximum of 35.3 MPa was recorded when the vertical displacement of the cylinder reached 30 mm). At these stress levels, the slope of the stress-strain curve for the MJF-printed PA12 was significantly lower than the slope within the range of 0.05% to 0.25% strain, which was considered for determining the elastic moduli used in the constitutive model selected to characterise the mechanical behaviour of the printed material. The constitutive model used to characterise the behaviour of the MJF-printed PA12 took no account of the progressive decrease in the stiffness of the material as the stress/strain level increased. This omission could also partially explain the difference observed between the results of the mechanical tests on the printed handles and the predictions resulting from the numerical model that was developed to simulate the mechanical response of those handles during the tests, with the latter overestimating the stiffness of the printed handles.

Using a constitutive model that assumes a transversely isotropic behaviour under linear elastic deformation for predicting the mechanical response of MJF-printed PA12 parts in numerical calculations could be a reasonable approach, provided that the stress levels of the material remain within the linear range. Otherwise, the numerical model might overestimate the stiffness of the printed material. Nevertheless, in the case of polymer materials, it is not always easy to define the separation point between the elastic and plastic fields, as their stress-strain curves can exhibit a smooth transition from the initial elastic response to irreversible plastic deformation [7,11]. This behaviour can be observed in the stress-strain curves of the MJF-printed PA12 shown in Figure 11 and Figure 23. Some authors, such as Calignano et al. [7], defined an arbitrary offset point to determine the yield point of the MJF-printed PA12, using the stress value at 1% as the reference value for the yield strength of the material. However, the appropriate value to be used as a reference for determining the yield strength of the MJF-printed PA12 will generally depend on the intended application for the printed part.

More complex constitutive models can also be used to predict the mechanical behaviour of MJF-printed PA12 parts through numerical simulations. Some authors, such as Shen et al. [34], Abueidda et al. [35], Schneider and Kumar [36], and Chen et al. [21], employed advanced constitutive models to investigate the deformation of powder-based 3D-printed polymers (including MJF-printed PA12), which can even account for viscoelastic and viscoplastic effects. However, integrating those constitutive models into numerical simulations usually demands significant effort, as user-defined material subroutines need to be employed for that purpose.

## 4. Conclusions

The aim of this study was to assess the suitability of a material constitutive model based on a transversely isotropic behaviour under linear elastic deformation for predicting the mechanical response of MJF-printed PA12 parts through numerical simulations.

First of all, assuming that the MJF-printed PA12 behaves as a transversely isotropic material, the decomposition of its overall three-dimensional behaviour into two simpler plane stress cases helped to determine, by means of uniaxial tensile and shear tests on samples printed in different build orientations, the elastic parameters that are required to fully define the flexibility matrix that characterises that behaviour within the linear elastic range. The analysis of those test results led to some conclusions on the anisotropy of the mechanical properties of the MJF-printed PA12:MJF-printed PA12 showed superior tensile properties along the vertical growing direction compared to the horizontal printing plane. Specifically, the tensile strength and elastic modulus values were approximately 16% higher along the vertical direction. The Poisson’s ratio values were relatively consistent on both the plane of isotropy of the material and the planes perpendicular to it. The notable variability in the results for the elongation at break complicated any definitive conclusions on the influence of the build orientation on that same parameter.The anisotropy of the shear properties of the MJF-printed PA12 was significantly lower compared to its tensile properties. Shear modulus values were nearly identical on both the plane of isotropy of the material and the vertical planes. Moreover, slightly higher shear strength values (+6.7%) were observed on the vertical planes.

Very similar average values for both vertical orientations were observed when comparing the results from uniaxial tensile and shear tests on samples printed along theoretically equivalent build orientations. However, regarding the printing plane, there was a more significant discrepancy in average values between theoretically equivalent build orientations, particularly for the tensile strength (difference of 17.42%) and elastic modulus (difference of 15.05%). Taking all of that into account, assuming a transversely isotropic material behaviour for the MJF-printed PA12 can be considered to be a reasonably valid approach.

The results of both mechanical tests conducted on handles printed along three different build orientations and a numerical model developed to simulate those tests were compared to assess the applicability of the material constitutive model. Qualitatively, the results obtained from the numerical model closely resembled the results of the mechanical tests on the printed handles, revealing a remarkably consistent behaviour for the three handles. Conversely, quantitatively speaking, the numerical model yielded stiffness values for the handles that were 26% to 32% higher than those obtained from the mechanical tests.

That difference can be attributed to two main reasons:The elastic moduli values used in the numerical model and obtained from uniaxial tensile tests on the printed samples were found to be significantly higher than those from uniaxial tensile tests on samples taken from the printed handles.The progressive reduction in material stiffness as the stress/strain increases was not considered in the material constitutive model employed in the numerical simulation to describe the mechanical behaviour of the MJF-printed PA12, leading to an overestimation of the stiffness of the printed material.

Using a constitutive model that assumes a transversely isotropic behaviour under linear elastic deformation to predict the mechanical response of MJF-printed PA12 parts through numerical simulations could be a feasible option in cases where the stress levels of the material remain within the linear range. However, determining the yield point in polymer materials is not always straightforward. More sophisticated and realistic constitutive models can also be employed to predict the mechanical behaviour of MJF-printed PA12 components in numerical simulations, although it will generally result in higher costs.

## Figures and Tables

**Figure 1 polymers-16-00056-f001:**
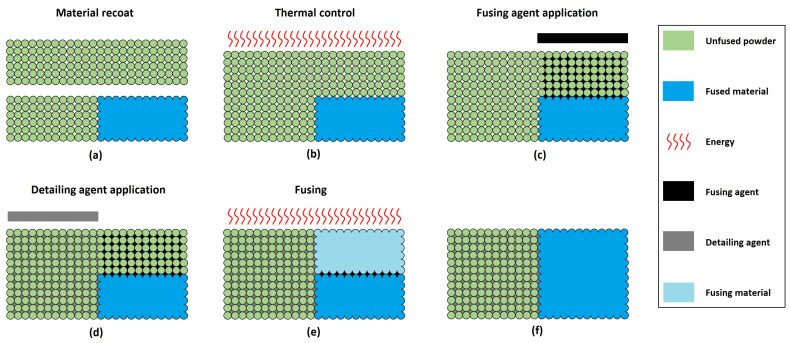
Schematic of Hewlett-Packard (HP) Inc. Multi Jet Fusion (MJF) printing process with material recoat (**a**), thermal control (**b**), application of the fusing (**c**) and detailing (**d**) agents, fusing (**e**) and end of cycle (**f**), adapted from [6].

**Figure 2 polymers-16-00056-f002:**
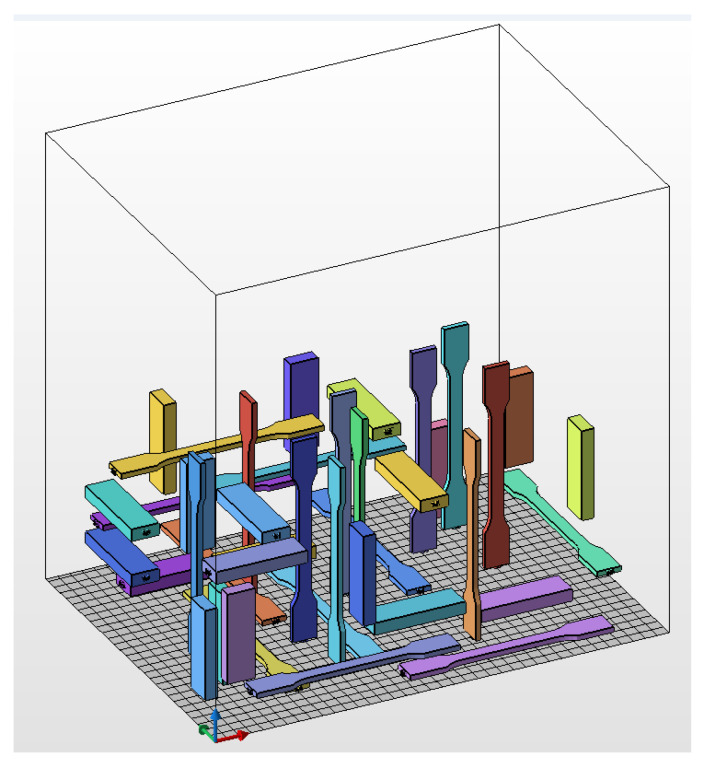
Sample layout for uniaxial tensile and shear tests within the build chamber of the printer.

**Figure 3 polymers-16-00056-f003:**
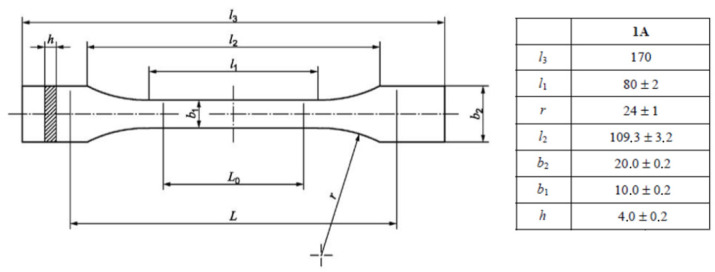
Geometry and dimensions (in mm) of type 1A samples, according to ISO 527-2 [26].

**Figure 4 polymers-16-00056-f004:**
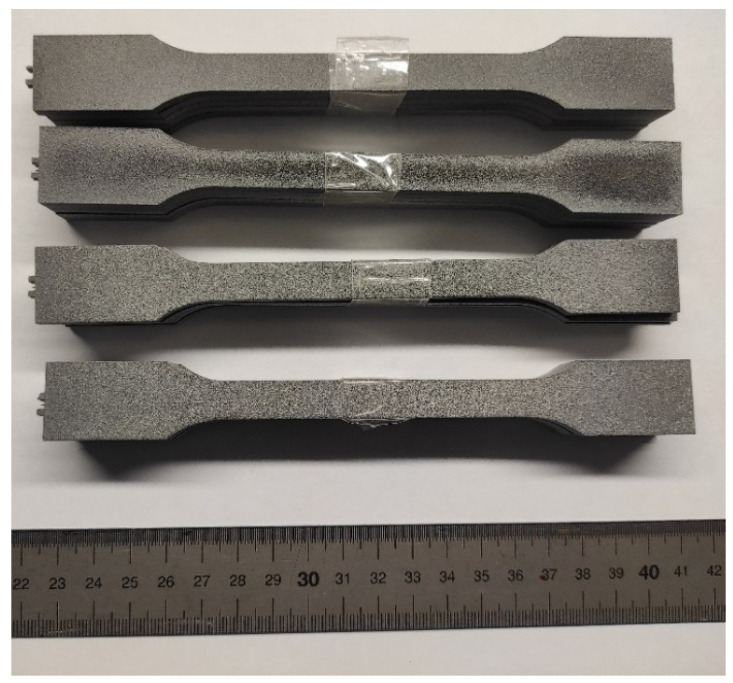
Printed samples for testing under uniaxial tensile loads.

**Figure 5 polymers-16-00056-f005:**
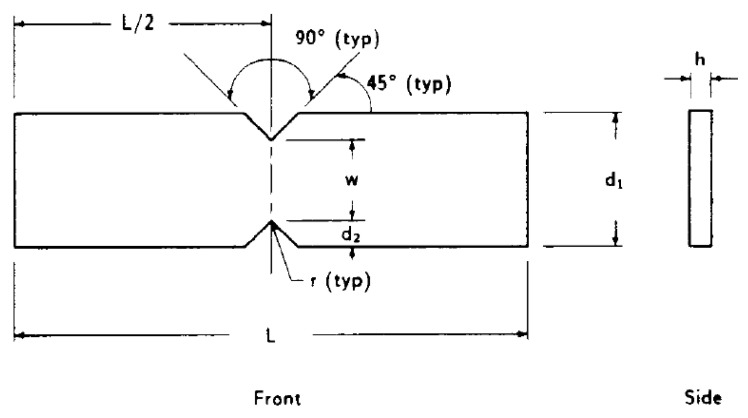
Geometry and dimensions of shear samples, according to ASTM D5379/D5379M [31].

**Figure 6 polymers-16-00056-f006:**
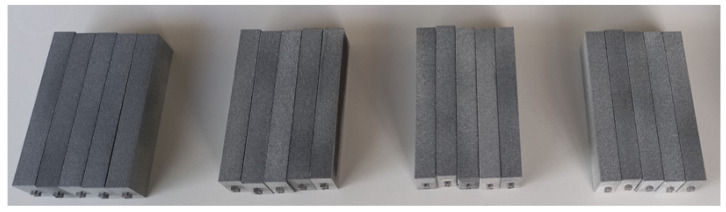
Oversized printed shear samples before machining and notching operations.

**Figure 7 polymers-16-00056-f007:**
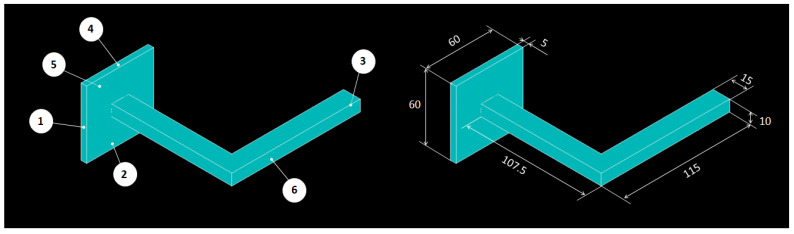
Geometry and dimensions of the handles used in this work, adapted from [32].

**Figure 8 polymers-16-00056-f008:**
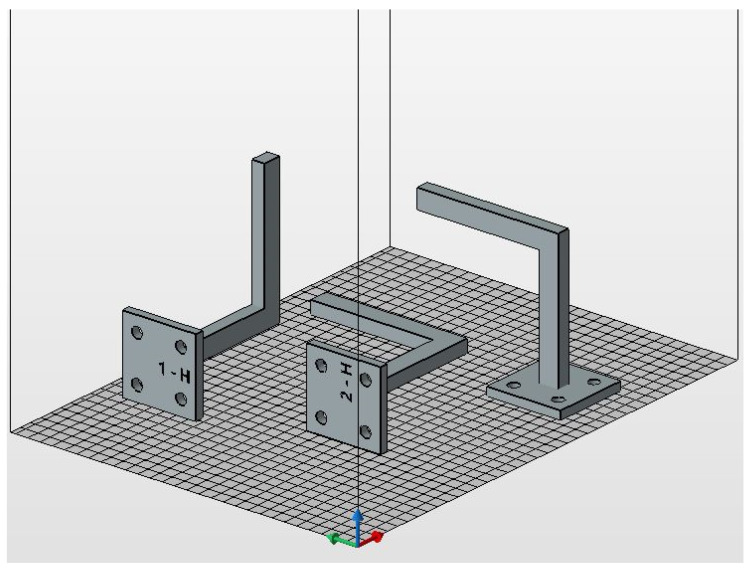
Layout of the handles for mechanical testing within the build chamber of the printer.

**Figure 9 polymers-16-00056-f009:**
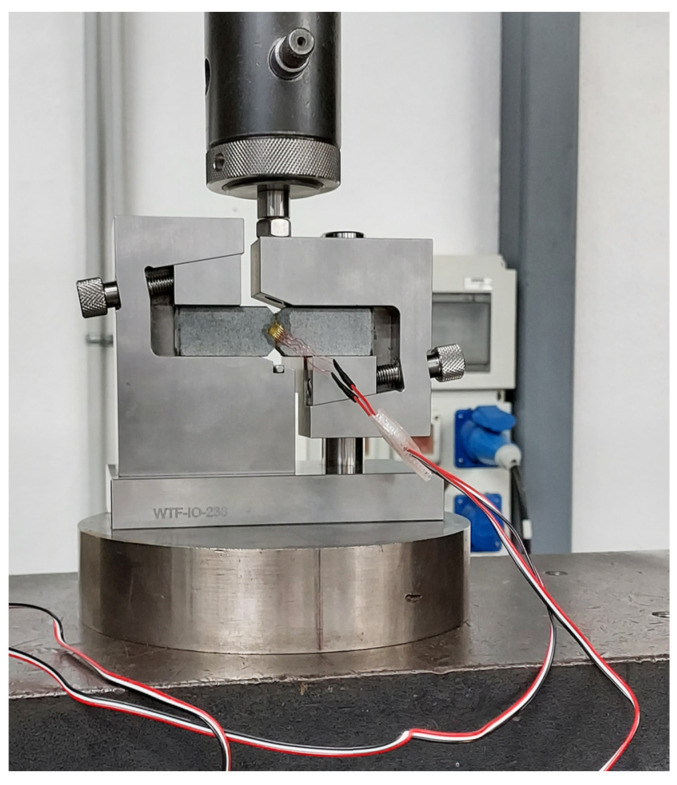
Shear sample mounted on the testing device.

**Figure 10 polymers-16-00056-f010:**
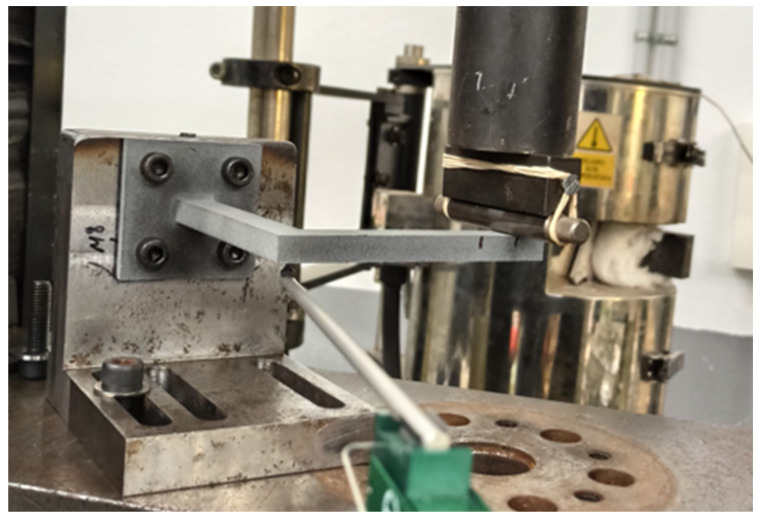
Experimental setup for the mechanical test on printed handles.

**Figure 11 polymers-16-00056-f011:**
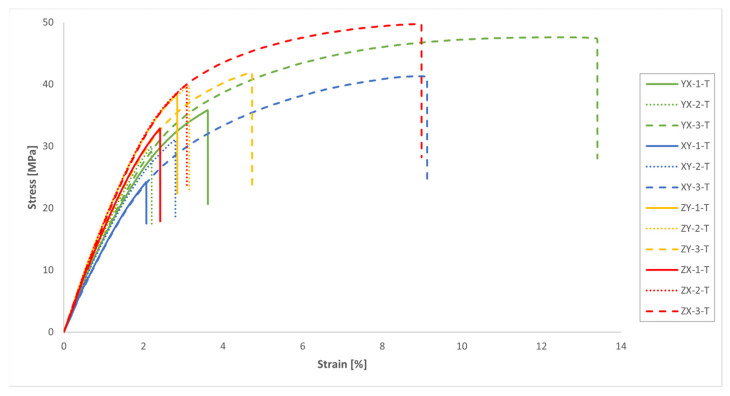
Tensile stress-strain curves of the MJF-printed PA12 samples (engineering values).

**Figure 12 polymers-16-00056-f012:**
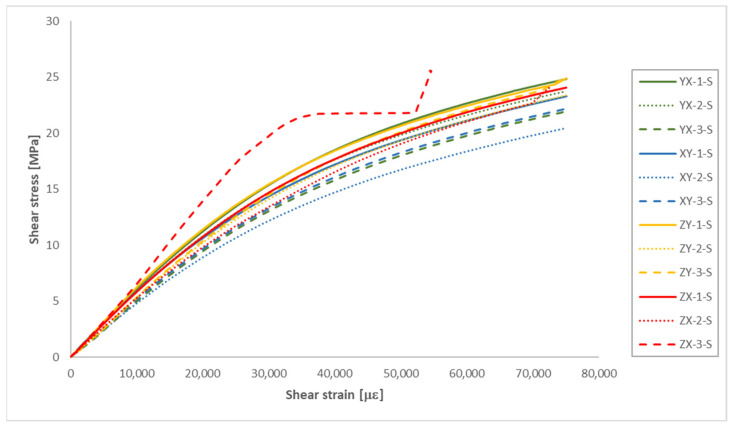
Shear stress-shear strain curves of the MJF-printed PA12 samples (engineering values).

**Figure 13 polymers-16-00056-f013:**
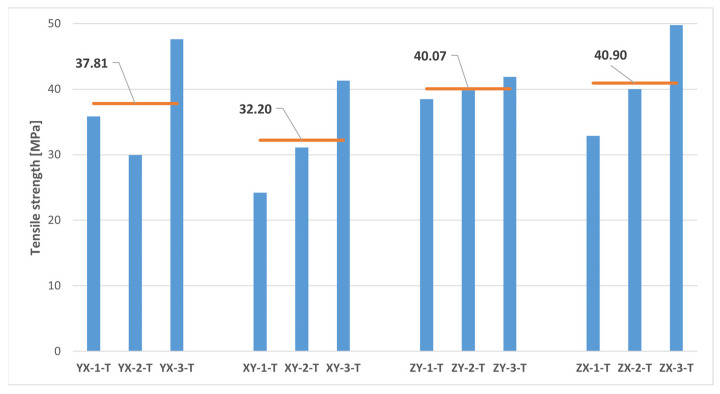
Tensile strength of printed samples tested under uniaxial tensile loads. The horizontal orange lines represent the average tensile strength value for each build orientation.

**Figure 14 polymers-16-00056-f014:**
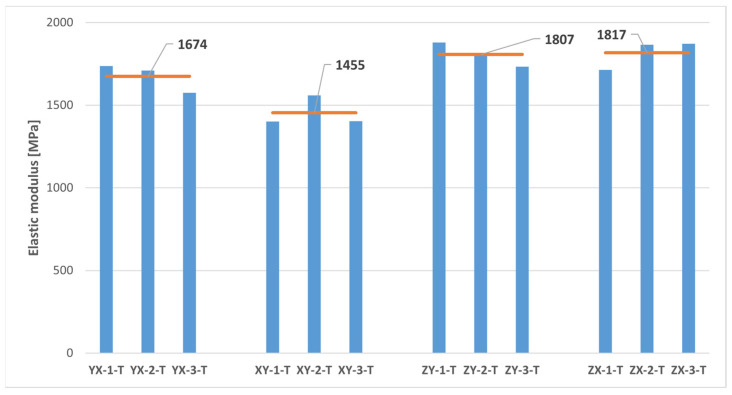
Elastic modulus of printed samples tested under uniaxial tensile loads. The horizontal orange lines represent the average elastic modulus value for each build orientation.

**Figure 15 polymers-16-00056-f015:**
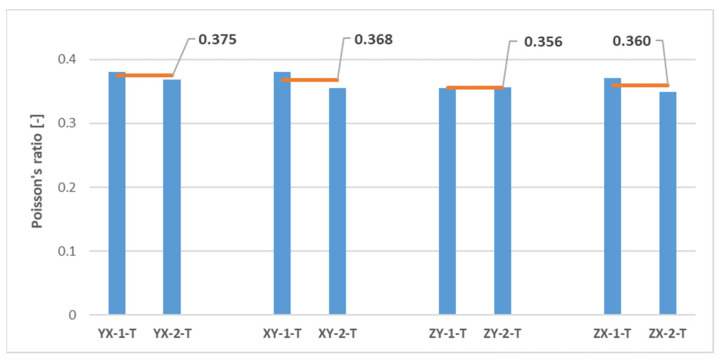
Poisson’s ratio of printed samples tested under uniaxial tensile loads. The horizontal orange lines represent the average Poisson’s ratio value for each build orientation.

**Figure 16 polymers-16-00056-f016:**
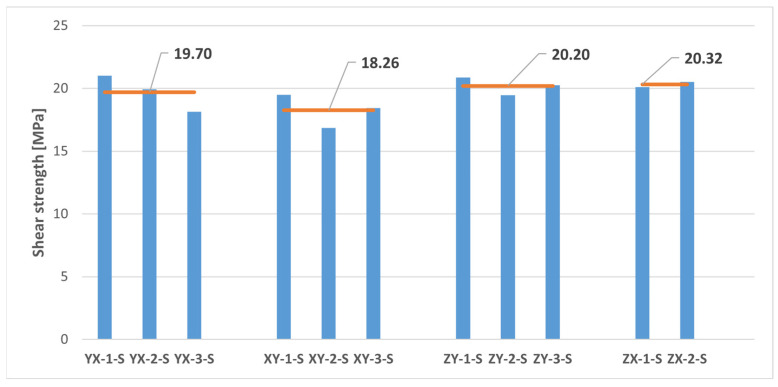
The shear strength of printed samples tested under shear loads. The horizontal orange lines represent the average shear strength value for each build orientation.

**Figure 17 polymers-16-00056-f017:**
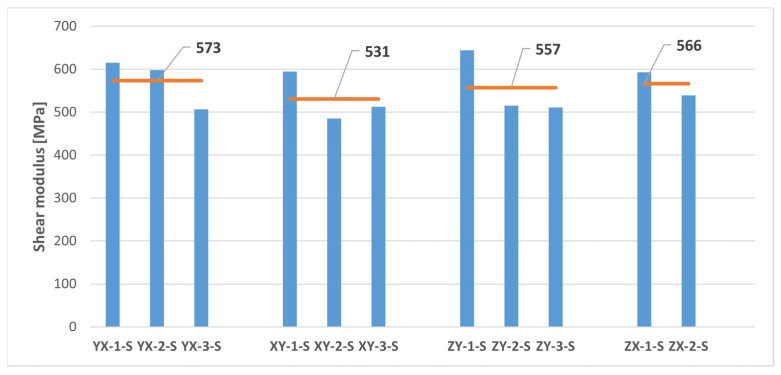
Shear modulus of printed samples tested under shear loads. The horizontal orange lines represent the average shear modulus value for each build orientation.

**Figure 18 polymers-16-00056-f018:**
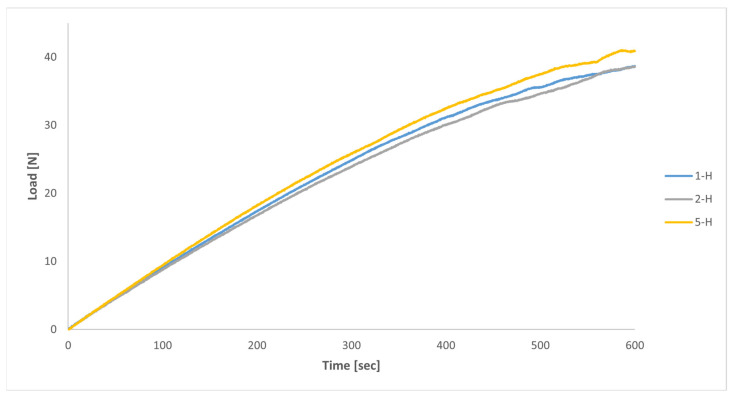
Load-time curves from mechanical tests on printed handles (strain rate = 5 mm/min).

**Figure 19 polymers-16-00056-f019:**
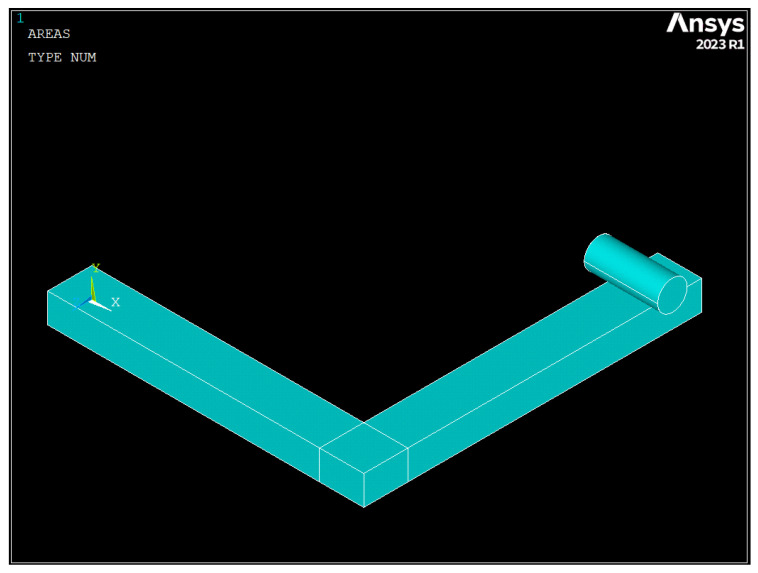
Geometry of the numerical model (ANSYS 2023R1).

**Figure 20 polymers-16-00056-f020:**
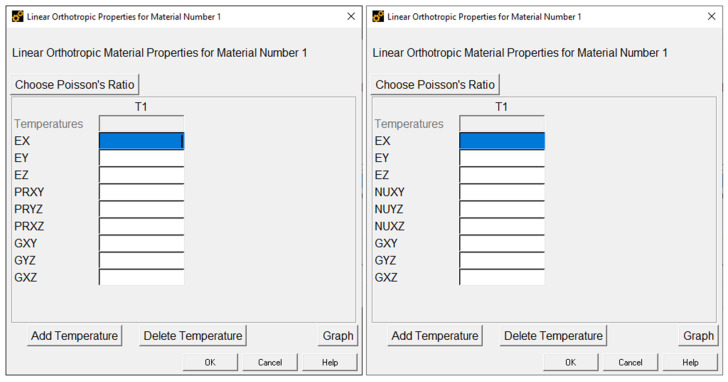
The dialogue box to input the properties of a linear orthotopic material model (ANSYS 2023R1).

**Figure 21 polymers-16-00056-f021:**
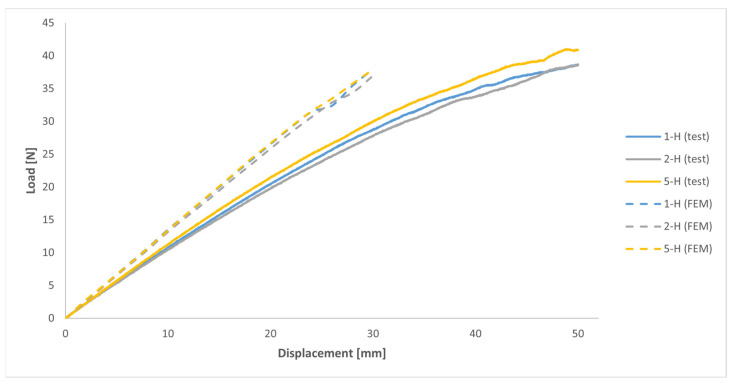
Load-displacement (at 15 mm from the edge of the handle) curves from numerical analyses and mechanical tests on handles.

**Figure 22 polymers-16-00056-f022:**
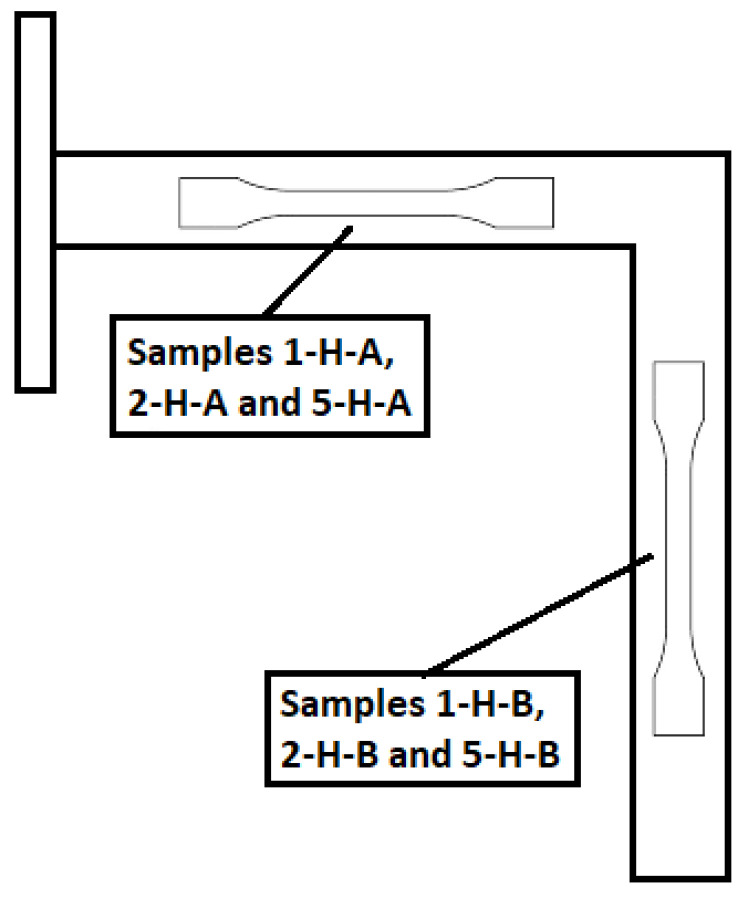
Tensile samples extracted from the printed handles.

**Figure 23 polymers-16-00056-f023:**
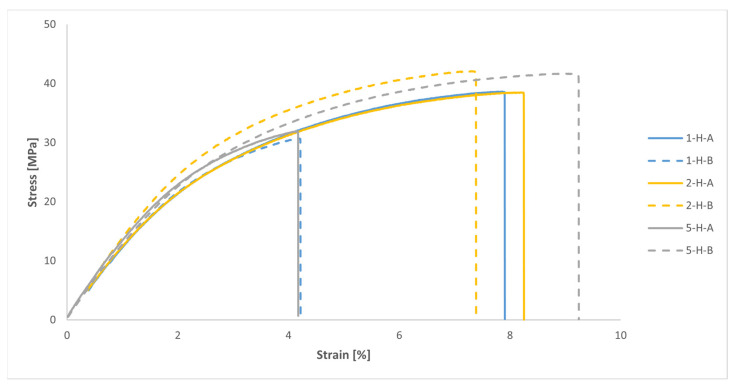
Tensile stress-strain curves of samples extracted from the printed handles (engineering values).

**Figure 24 polymers-16-00056-f024:**
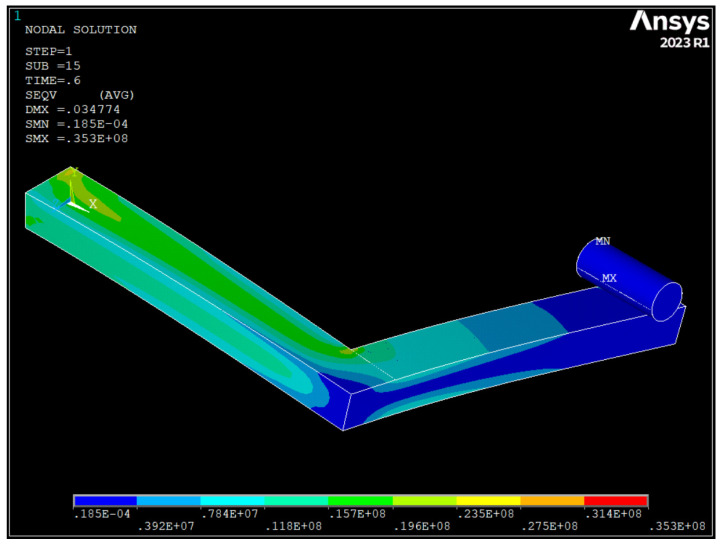
Results of the numerical model for handle 5-H. Von Mises equivalent stress (in Pa) resulting from a cylinder displacement of 30 mm.

**Table 1 polymers-16-00056-t001:** Works available in the literature regarding the anisotropy of the mechanical properties of MJF-printed PA12.

Reference	Type of Tests/Build Orientations/Standard	Printer	Print Mode
O’Connor et al. [13]	Tensile/XYZ (X), YZX (Y) and ZYX (Z)/ISO 527 [26]	HP Jet Fusion 4200	Balanced
	Flexural/XYZ (X), YZX (Y) and ZYX (Z)/ISO 178 [27]		
O’Connor and Dowling [16]	Tensile/XYZ (X), YZX (Y) and ZYX (Z)/ASTM D638 [28]	HP Jet Fusion 4200	Balanced
	Flexural/XYZ (X), YZX (Y) and ZYX (Z)/ASTM D790 [29]		
Morales-Planas et al. [17]	Tensile/XY, YZ and ZX/ASTM D638	HP Jet Fusion 4200	Balanced
Palma et al. [5]	Tensile/Vertical and horizontal/ASTM D638	-	-
Riedelbauch et al. [15]	Tensile/YXZ and ZXY/ISO 527	HP Jet Fusion 4200	Balanced
Galati et al. [9]	Tensile/XY and Z/ISO 527	HP Jet Fusion 4200	-
	Tensile/XY, XY-50°, Z and Z-50°/-		
Sillani et al. [18]	Tensile/X, Y and Z/ISO 527	HP Jet Fusion 4200	-
Mehdipour et al. [19]	Tensile/Flatwise, edgewise and upright/ISO 527	HP Jet Fusion 4200	Balanced
Cai et al. [8]	Tensile/X, Y and Z/ASTM D638	HP Jet Fusion 4200	Balanced
	Flexural/X, Y and Z/ISO 178		
Rosso et al. [20]	Tensile/Z/ISO 527	HP Jet Fusion 4200	Balanced
Calignano et al. [7]	Tensile/fx, fd, fy, vx, vd, vy, zx, zd and zy/ISO 527	HP Jet Fusion 4200	Balanced
Chen et al. [21]	Tensile/XY_0°, XY_90° and Z/ASTM D638	HP Jet Fusion 5200	Balanced
	Shear/XY_0° and Z/ASTM B831 [30]		
Chen et al. [22]	Tensile/Vertical and horizontal/-	HP Jet Fusion 5200	-
Osswald et al. [23]	Tensile/Vertical and horizontal/-	HP Jet Fusion 4200	Balanced
	Compression/Vertical and horizontal/-		
	Torsion/Vertical, horizontal and 45°/-		
	Torsion + axial/Vertical, horizontal and 45°/-		
Abdallah et al. [24]	Tensile/0° and 25°/ASTM D638	-	-
Koh et al. [12]	Tensile/X, X45 and Z/ASTM D638	HP Jet Fusion 5200	-
HP Inc. [14]	Tensile/XY, YX and Z/ASTM D638	HP Jet Fusion 4200	Balanced

**Table 2 polymers-16-00056-t002:** HP Jet Fusion 4200—Polyamide 12 (PA12): Technical characteristics.

Powder:	PA12
Average particle size [µm]:	60
Bulk density of powder [g/cm^3^]:	0.425
Density of parts [g/cm^3^]:	1.01
Powder melting point [°C]:	187
Layer thickness [µm]:	80
Mixing ratio of virgin/recycled powder:	20:80
Build volume [mm^3^]:	380 × 254 × 380

**Table 3 polymers-16-00056-t003:** Results of the uniaxial tensile tests on the MJF-printed PA12 samples.

Sample	Width [mm]	Thickness [mm]	Maximum Load [N]	Tensile Strength [MPa]	Elastic Modulus [GPa]	Elongation at Break [%]	Poisson’s Ratio [-]
YX-1-T	9.95	4.08	1455	35.84	1.737	3.61	0.381
YX-2-T	10.03	4.06	1221	29.98	1.709	2.20	0.369
YX-3-T	9.97	4.06	1927	47.61	1.576	13.36	-
XY-1-T	9.84	4.13	985	24.20	1.401	2.07	0.381
XY-2-T	9.88	4.10	1259	31.10	1.559	2.80	0.355
XY-3-T	9.90	4.14	1695	41.30	1.404	9.12	-
YX + XY		Mean value	1424 ± 343.09	35.00 ± 8.44	1.564 ± 0.144	5.53 ± 4.66	0.372 ± 0.012
ZY-1-T	10.05	4.00	1550	38.50	1.879	2.85	0.355
ZY-2-T	10.10	4.03	1619	39.80	1.809	3.14	0.356
ZY-3-T	10.02	4.00	1679	41.90	1.734	4.70	-
ZX-1-T	10.00	3.90	1283	32.90	1.714	2.41	0.371
ZX-2-T	10.10	4.06	1640	40.00	1.866	3.08	0.349
ZX-3-T	10.15	4.07	2056	49.80	1.872	8.96	-
ZY + ZX		Mean value	1638 ± 249.21	40.48 ± 5.49	1.812 ± 0.073	4.19 ± 2.46	0.358 ± 0.010

**Table 4 polymers-16-00056-t004:** Tensile properties of MJF-printed PA12 from current literature.

Reference	Tensile Strength [MPa]	Elastic Modulus [GPa]	Elongation at Break [%]
O’Connor et al. [13] O’Connor and Dowling [16]	X = 47 ± 0.9 Y = 48 ± 0.8 Z = 49 ± 0.6	X = 1.242 ± 0.028 Y = 1.147 ± 0.040 Z = 1.246 ± 0.037	X = 19 ± 2.8 Y = 27 ± 1.2 Z = 16 ± 1.9
Morales-Planas et al. [17]	XY = 47.9 ÷ 51.6 YZ = 45.6 ÷ 52.1 ZX = 50.9 ÷ 57.4	XY = 3.525 ÷ 4.202 YZ = 3.767 ÷ 4.321 ZX = 4.106 ÷ 4.409	XY = 2.5 ÷ 4.1 YZ = 2.0 ÷ 2.5 ZX = 2.1 ÷ 4.8
Palma et al. [5]	H = 45.15 V = 47.77	-	H = 23.2 V = 17.4
Riedelbauch et al. [15]	YXZ = 46.7 ZXY = 52.3	YXZ = 1.439 ZXY = 1.580	YXZ = 13.8 ZXY = 12.5
Galati et al. [9]	XY = 36 Z = 39	-	XY = 25 Z = 18
Sillani et al. [18]	X = 45.8 ± 3.5 Y = 47.9 ± 0.9 Z = 53.7 ± 1.1	X = 1.128 ± 0.068 Y = 1.204 ± 0.084 Z = 1.337 ± 0.098	X = 11.2 ± 1.8 Y = 13.2 ± 1.5 Z = 11.4 ± 1.3
Mehdipour et al. [19] *	Flatwise = 34.39 ± 1.71 Edgewise = 44.07 ± 0.79 Upright = 42.79 ± 0.38	Flatwise = 1.063 ± 0.025 Edgewise = 1.435 ± 0.024 Upright = 1.495 ± 0.039	Flatwise = 17.19 ± 1.36 Edgewise = 16.39 ± 0.28 Upright = 11.98 ± 1.38
Cai et al. [8]	X = 48.7 ± 0.8 Y = 44.5 ± 0.7 Z = 49.6 ± 1.2	X = 1.369 ± 0.025 Y = 1.369 ± 0.069 Z = 1.669 ± 0.067	X = 27.4 ± 2.2 Y = 15.9 ± 1.1 Z = 14.8 ± 0.3
Rosso et al. [20]	Z = 45.6 ± 0.4	Z = 1.53 ± 0.06	Z = 30.0 ± 4.9
Calignano et al. [7]	fx = 35.4 ± 2.6 fd = 34.0 ± 2.6 fy = 35.2 ± 2.0 vx = 38.2 ± 1.9 vd = 30.5 ± 4.6 vy = 35.2 ± 0.6 zx = 38.4 ± 3.3 zd = 39.8 ± 0.7 zy = 36.8 ± 2.5	fx = 1.223 ± 0.157 fd = 1.170 ± 0.136 fy = 1.286 ± 0.029 vx = 1.326 ± 0.070 vd = 0.974 ± 0.087 vy = 1.337 ± 0.083 zx = 1.205 ± 0.536 zd = 1.499 ± 0.291 zy = 1.513 ± 0.296	fx = 21.5 ± 7.3 fd = 13.7 ± 1.6 fy = 15.9 ± 3.2 vx = 25.3 ± 4.2 vd = 15.1 ± 8.1 vy = 13.2 ± 2.0 zx = 11.2 ± 8.4 zd = 18.5 ± 1.7 zy = 18.0 ± 1.1
Chen et al. [22]	H = 45.8 ± 0.5 V = 45.7 ± 0.7	H = 1.436 ± 0.043 V = 1.561 ± 0.031	H = 29.3 ± 3.8 V = 10.7 ± 0.2
Osswald et al. [23]	H = 41.24 ± 1.18 V = 48.97 ± 1.01	-	-
HP Inc. [14]	XY = 50 Z = 50	XY = 1.7 Z = 1.9	XY = 17 Z = 9
Current work	YX + XY = 35.00 ± 8.44 ZY + ZX = 40.48 ± 5.49	YX + XY = 1.564 ± 0.144 ZY + ZX = 1.812 ± 0.073	YX + XY = 5.53 ± 4.66 ZY + ZX = 4.19 ± 2.46

* samples tested at 5 mm/min.

**Table 5 polymers-16-00056-t005:** Results of the shear tests on the MJF-printed PA12 samples.

Sample	Width [mm]	Thickness [mm]	Load ε=5% [N]	Shear Strength [MPa]	Shear Modulus [GPa]
YX-1-S	12.00	10.00	2523	21.03	0.615
YX-2-S	12.00	10.00	2392	19.93	0.598
YX-3-S	12.00	10.10	2198	18.14	0.507
XY-1-S	12.00	10.00	2339	19.49	0.595
XY-2-S	12.00	10.00	2023	16.86	0.485
XY-3-S	12.00	10.00	2212	18.43	0.513
YX + XY		Mean value	2281 ± 174.65	18.98 ± 1.47	0.552 ± 0.057
ZY-1-S	12.00	10.00	2505	20.88	0.644
ZY-2-S	12.00	10.00	2336	19.47	0.515
ZY-3-S	12.00	10.00	2430	20.25	0.511
ZX-1-S	12.00	9.90	2389	20.11	0.593
ZX-2-S	12.00	9.95	2451	20.53	0.539
ZY + ZX		Mean value	2422 ± 63.69	20.25 ± 0.52	0.560 ± 0.057

**Table 6 polymers-16-00056-t006:** Properties of the MJF-printed PA12 used in the numerical model for each of the three build orientations considered for the printed handles.

	1-H	2-H	5-H
*EX* (GPa)	1.564	1.564	1.812
*EY* (GPa)	1.564	1.812	1.564
*EZ* (GPa)	1.812	1.564	1.564
*PRXY* (-)	0.372	0.309 *	0.309 *
*PRYZ* (-)	0.309 *	0.309 *	0.372
*PRXZ* (-)	0.309 *	0.372	0.309 *
*GXY* (GPa)	0.552	0.560	0.560
*GYZ* (GPa)	0.560	0.560	0.552
*GXZ* (GPa)	0.560	0.552	0.560

* 0.309 = 1.564 × 0.358/1.812.

**Table 7 polymers-16-00056-t007:** Uniaxial tensile test results of samples extracted from the printed handles.

Sample	Width [mm]	Thickness [mm]	Maximum Load [N]	Tensile Strength [MPa]	Elastic Modulus [GPa]	Elongation at Break [%]
1-H-A	5.04	3.95	769	38.63	1.317	7.90
2-H-A	5.03	3.97	769	38.51	1.217	8.20
2-H-B	4.93	3.97	824	42.10	1.436	7.40
5-H-B	4.95	3.98	821	41.67	1.339	9.20
		Mean value	796 ± 30.91	40.23 ± 1.92	1.327 ± 0.090	8.18 ± 0.76
YX + XY (Table 3)				35.00 ± 8.44	1.564 ± 0.144	5.53 ± 4.66
1-H-B	4.98	3.98	609	30.73	1.256	4.20
5-H-A	4.98	3.98	633	31.94	1.430	4.20
		Mean value	621 ± 16.97	31.34 ± 0.56	1.343 ± 0.123	4.20 ± 0.00
ZY + ZX (Table 3)				40.48 ± 5.49	1.812 ± 0.073	4.19 ± 2.46

## Data Availability

The data presented in this study are available on request from the corresponding author.

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
