# Peer review of "Applicability of a Material Constitutive Model Based on a Transversely Isotropic Behaviour for the Prediction of the Mechanical Performance of Multi Jet Fusion Printed Polyamide 12 Parts"

_polymers, 2023, doi:10.3390/polym16010056_

Round 1

Reviewer 1 Report

Comments and Suggestions for Authors

The article explores the relatively new technology of 3D printing using a commercially available HP printer. This technology is of great interest for practical implementation due to its printing speed and resolution. The article may be accepted for publication after minor revisions.

Please add a direction or number to diagram 1. If the diagram is borrowed from another article, you must obtain permission to use or provide a link.

The introduction was very long. I shortened it by adding a comparison table with parameters specified in the standard and the derivation method. It is possible to add a table after performing mechanical tests to show comparative results.

Why were printing handles chosen? The article provides a significant amount of data on mechanical tests, which are the most relevant. It is better to present a topologically conservative bracket geometry or benchmark models.

It is preferable to display the average and measurement error in Tables 2 and 3, which will make them more readable.

In the mechanical test graphs, it is necessary to indicate whether these are engineering or true values.

Add the measurement error to graphs 13-17.

The use of finite element analysis may contain redundant information. The article is overloaded with experimental data.

Author Response

Dear reviewer.

First of all, many thanks for your valuable comments. Please, find below my answers to them. Best regards.

Comment: The article explores the relatively new technology of 3D printing using a commercially available HP printer. This technology is of great interest for practical implementation due to its printing speed and resolution. The article may be accepted for publication after minor revisions.

Answer: No answer is required.

Comment: Please add a direction or number to diagram 1. If the diagram is borrowed from another article, you must obtain permission to use or provide a link.

Answer: Figure 1 has been updated (see Figure 1 in the attached document). Moreover, the manuscript has been slightly modified to improve the explanation of the MJF printing process. Specifically, the different phases of that process have been related to the corresponding subfigures in Figure 1, which should help clarify the process for the reader (see lines 54 to 69 in the attached document). Figure 1 is an original diagram, so no permission is needed.

Comment: The introduction was very long. I shortened it by adding a comparison table with parameters specified in the standard and the derivation method. It is possible to add a table after performing mechanical tests to show comparative results.

Answer: The part of the introduction regarding the works that analyzed the anisotropy of the mechanical properties of MJF-printed PA12 has been shortened (see lines 81 to 210 in the attached document), and a table describing the type of tests carried out, the printer model and the print mode employed for sample fabrication, and the build orientations considered during the sample printing process in those works has been included (see Table 1 in the attached document). A table summarizing published data regarding the tensile properties of MJF-printed PA12 has been added to the manuscript (see Table 4 in the attached document).

Comment: Why were printing handles chosen? The article provides a significant amount of data on mechanical tests, which are the most relevant. It is better to present a topologically conservative bracket geometry or benchmark models.

Answer: Printed standardized samples were manufactured and tested under uniaxial tensile and shear loads to determine the elastic properties and evaluate the anisotropic behaviour of MFJ-printed PA12. In those cases, due to the geometry of the samples and the particularities of the tests, printed samples experienced simple stress states during testing.

Nevertheless, real parts are usually subjected to actions of different nature that generate more complex stress states on the material. Then, in order to assess the applicability of the selected material constitutive model (which was built based on the elastic properties obtained from uniaxial tensile and shear tests) to predict the mechanical performance of real MJF-printed PA12 parts in numerical simulations, more complex handle-shaped were also analyzed. On the one hand, printed handles were manufactured and tested. On the other hand, a numerical model using the selected material constitutive model was developed to simulate the mechanical tests on the printed handles. This way, the comparison between the results from both mechanical tests on printed handles and the numerical model helped to establish conclusions regarding the applicability of the selected material constitutive model to predict the mechanical performance of real MJF-printed PA12 parts in numerical simulations.

Comment: It is preferable to display the average and measurement error in Tables 2 and 3, which will make them more readable.

Answer: Those tables have been updated, eliminating the rows corresponding to the standard deviation and coefficient of variation. Now, in those tables, average values are shown in the form “average value ± standard deviation” (see Tables 3, 5 and 7 in the attached document).

Comment: In the mechanical test graphs, it is necessary to indicate whether these are engineering or true values.

Answer: The test “(engineering values)” has been added at the end of the caption in Figures 11, 12, and 23 (see attached document).

Comment: Add the measurement error to graphs 13-17.

Answer: The information shown in graphs 13-17 was obtained from individual tests performed on different specimens. The bars of those graphs show the individual mechanical properties of each of the specimens tested, while the horizontal lines represent the average values for each build orientation. Being information from individual tests, the bar should not have a measurement error and the authors deemed not necessary to represent the error induced by obtaining the average shown using the orange lines.

Comment: The use of finite element analysis may contain redundant information. The article is overloaded with experimental data.

Answer: The finite element analysis was necessary because the results from both the numerical model and the mechanical tests on the printed handles needed to be compared in order to draw conclusions regarding the applicability of the selected material constitutive model to predict the mechanical performance of real MJF-printed PA12 parts in numerical simulations.

Reviewer 2 Report

Comments and Suggestions for Authors

It seems that the simulation was done linear (small displacement/ linear material) No information about the type of simulation (implicit, explicit, transient, time steping) in provided. i would recomend to not use mises stress. Please explain why this was used and not a failure criteria like tsai/hill, hoffman, tsai wu... 

Author Response

Dear reviewer.

First of all, many thanks for your valuable comments. Please, find below my answers to them. Best regards.

Comment: It seems that the simulation was done linear (small displacement/linear material).

Answer: As indicated in the manuscript, a constitutive model based on a linear material behaviour was employed for the characterization of the MJF-printed PA12 in the numerical model. However, the presence of a frictionless contact between the cylinder on which the vertical displacement was imposed and the printed handle required a nonlinear analysis. The whole analysis was decomposed into 25 substeps, equivalent to a vertical displacement of 2 mm per substep. Additional information regarding the type of analysis used has been included in the manuscript (see lines 587 to 592 in the attached document).

Comment: No information about the type of simulation (implicit, explicit, transient, time stepping) is provided.

Answer: An implicit solver was used for the resolution of the numerical model. Additional information regarding the type of solver used has been included in the manuscript (see lines 587 to 592 in the attached document).

Comment: I would recommend to not use von mises stress. Please explain why this was used and not a failure criteria like Tsai/hill, Hoffman, Tsai Wu...

Answer: Regarding the use of the Von Mises stress in Figure 24, the purpose of that figure was not to indicate whether the printed handle was close or far from its load-bearing capacity. If that had been the case, as you rightly point out, a failure criterion for orthotropic materials, such as Tsai-Hill, Hoffman, or Tsai-Wu, should have been employed. Here, Figure 24 was included just to qualitatively show that, during the simulation of the mechanical test on the printed handles, the stress level reached in the MJF-printed PA12 was very high, and at this stress level, the slope of the stress-strain curve for the MJF-printed PA12 was significantly lower than the slope within the range of 0.05% to 0.25% strain, which was considered for determining the elastic moduli used in the constitutive model selected to characterize the mechanical behaviour of the printed material. If any of the mentioned failure criteria (Tsai-Hill, Hoffman or Tsai-Wu) had been used in Figure 24, that figure would have shown a value between 0 and 1 for each element of the numerical model, making it harder the comparison between Figure 24 and Figures 11 and 23.
